# A Fifty-Year Experience of Groundwater Governance: The Case Study of Gakunan Council for Coordinated Groundwater Pumping, Fuji City, Shizuoka Prefecture, Japan

**Takahiro Endo**

College of Sustainable System Sciences, Osaka Prefecture University, Sakai, Osaka 599-8531, Japan; tte23042@osakafu-u.ac.jp

**Abstract:** Groundwater protection, which is effected by multiple actors at multiple levels using multiple instruments, is commonly termed "groundwater governance". Although the concept has attracted increasing attention since the 1990s, several of its associated measures remain to be fully implemented. Most are still inchoate strategies and improvement is expected to be a gradual, long-term process. The Gakunan Council for Coordinated Groundwater Pumping (CCGP), which was established, in 1967, in Fuji City in Japan's Shizuoka Prefecture, is an exceptional case. The Gakunan CCCP was created to deal with a common-pool resource problem where massive groundwater pumping caused seawater intrusion in the city's coastal area due to the low cost of extraction and incomplete groundwater ownership. The Gakunan CCCP succeeded in recovering elevation of groundwater tables by connecting efforts between the public and private sectors, including information sharing, legal authority to regulate groundwater, investment in alternative water supplies, internal subsidies between groundwater users, and charge for water disposal. Previous studies have iterated that the fostering of participation from various stakeholders and dividing labor between them appropriately are key elements of successful groundwater governance. This paper investigates these factors, explores the importance of the metagovernor as coordinator, and offers a fresh perspective on the significance of groundwater governance.

**Keywords:** groundwater governance; stakeholder; regulation; market; community; tragedy of the commons; excessive groundwater pumping; industrial waterworks

## 1. Introduction

According to Torfing et al., (2013) [1], governance can be defined as a process of steering society and the economy through collective action and in accordance with some common objectives. The achievement of social goals through the participation of stakeholders with various interests is currently attracting increased attention in the field of groundwater management [2–4].

As a concept, governance is neither inherently "good" nor "bad" [5]. The efficacy of groundwater governance depends on a range of natural and societal conditions, and therefore there is no universally applicable, "one-size-fits-all" solution [6,7], however, it is possible to extract generally applicable lessons from case studies by addressing the following questions: Under what conditions does a given groundwater-related policy tool work? What is the case for the engagement of various stakeholders? and How should the division of labor between public- and private-sector stakeholders be approached? [2,8].

The word "governance", in terms of its current application, is relatively new. It entered general use in the 1980s via studies on public administration and has gradually come to be regarded as a

key concept in groundwater issues since the 1990s [3]. As Foster and van der Gun (2016) and De Chaisemartin et al. (2017) observed, however, while theoretical analysis of groundwater governance has been conducted, the proposed measures remain to be implemented. Most are still inchoate strategies and improvement is expected to be a gradual and lengthy process [9,10].

In view of these circumstances, Japan's Council for Coordinated Groundwater Pumping (CCGP) may offer useful information as a case study for researchers and practitioners on groundwater governance. More than twenty such councils have been established at the local level throughout Japan. This paper focuses on one of the pioneering examples, Gakunan CCGP, which was established, in 1967, in Fuji City in Japan's Shizuoka Prefecture. As will be detailed below, this organization has successfully addressed the problem of excessive groundwater pumping, drawing on the support of various local public- and private-sector stakeholders, and applying a blend of different policy tools. Gakunan CCGP has implemented groundwater governance for 50 years, several decades in advance of the concept's current prominence, however, unfortunately, very few studies have examined Gakunan CCGP. Indeed, it would not be going too far to say that this case is unknown outside Japan.

This paper has two main objectives. First, to elucidate the history and function of Gakunan CCGP, using a framework developed in recent groundwater governance studies and, secondly, to extract generally applicable policy lessons from a case study of Gakunan CCGP. Previous studies have iterated that promoting the participation of various stakeholders and dividing labor appropriately between them are key elements in successful local groundwater governance [8,10,11]. This paper investigates these factors, explores the importance of the metagovernor as coordinator, and offers a different perspective on the significance of groundwater governance.

Conceptual analyses of governance and, more specifically, groundwater governance, are presented in the next section. The societal background of CCGP is explained, with references to Japan's groundwater-related institutions generally, in the third section. The history of Gakunan CCGP is described with reference to the four key elements of groundwater governance identified by Varady et al. (2016) [12] in the fourth section. General policy lessons for groundwater governance emerging from the Gakunan CCGP case study are presented in the fifth section. Finally, an overall summary is offered in the sixth section.

## 2. What is Groundwater Governance?

The term "governance" has been ascribed to several and various definitions, provoking the criticism that the concept is "notoriously slippery" [13]. Generally, it can refer to any pattern of rule, however, the nuances and intended meanings of words vary. For example, public sector reforms took place worldwide during the 1980s, frequently resulting in smaller governments, while the delivery of public services was delegated to private companies or nongovernmental organizations (NGOs). Proponents of neoliberalism who promoted this change called this new method of public service provision "governance." However, rational choice theories investigated the conditions under which people voluntarily complied with social norms, using the term "governance" to describe situations in which order was achieved without coercion from higher-level authorities [14].

Although the definition may differ across various academic fields, there is nonetheless a common element, i.e., the reconsideration of government functions or, put another way, the promotion of public policy by various stakeholders. Neoliberalists have directed attention toward the roles of private companies and NGOs, as well as those of the government, however, rational theorists have revised political science's traditional assertion that a fundamental public service, such as social order, cannot be provided without government, a social apparatus of coercion.

As environmental problems became a global issue, this way of thinking has gradually been applied to the field of natural resource management, now widely known as environmental governance. Although governance studies assume interplay between government and market NGOs, environmental governance studies focus on the interplay between government and the market community, the group that uses local resources. For example, Lemos and Agrawal (2006) defined environmental governance as

a process whereby actors, such as state agencies, market actors or communities, influence environmental actions and outcomes [15]. Young (2013) offered a similar view, portraying environmental governance as guiding and steering human behavior to avoid excessive exploitation of natural resources through the appropriate application of regulation, economic incentives, and social conventions [16].

Groundwater governance is a subdivision of environmental governance. Groundwater is a typical example of common-pool resources that have two key characteristics, difficulty in exclusion and rivalry in consumption. Difficulty in exclusion implies that controlling a range of beneficiaries through physical or institutional means may be prohibitively expensive. Rivalry in consumption indicates that an individual's consumption of the resource reduces the potential consumption of others. Natural resources, generally, tend to share these common-pool characteristics [17].

The variety of the resource's potential users is another characteristic of groundwater, which is frequently shared by individual users, private companies, irrigation districts, and public agencies for waterworks, who each exploit the resource at different levels and scales. Additionally, groundwater resources, especially those that are renewable, are not stock resources, but rather fluid ones, though the speed is significantly inferior to that of surface water. Owing to the difficulty in exclusion, groundwater often remains an open-access resource that is shared by an unspecified number of people. This mobility contributes to the transmission of negative externalities from one user to others. In such instances, users are likely to impose negative externalities on one another, resulting in excessive use of groundwater, and therefore the user heterogeneity raised transaction costs to reduce such externalities [18]. This phenomenon is widely known as the "tragedy of the commons" [19].

Thus, the coordination of various stakeholders at different levels is necessary to secure the benefits of groundwater and to prevent inefficient use of the resource. Representative methods of such coordination are classified into three groups, regulatory policy instruments, economic policy instruments, and voluntary policy instruments [20]. However, none of these offers a holistic solution; rather, an approach that combines different tools is required, depending on local circumstances [7]. How such combination was done in a local setting will be discussed in Section 5 with a groundwater governance matrix.

In summary, relying on the definition offered by Mukherji and Shah (2005), groundwater governance is a process of protecting groundwater that requires the implementation of multiple instruments by multiple actors from multiple levels [2].

## 3. Council for Coordinated Groundwater Pumping (CCGP)

### 3.1. CCGP in General

Groundwater and surface water are both components of the hydrological cycle and are often considered as a physical unit, with the exception of fossil groundwater. However, in a legal sense, they are generally regarded as separate resources. In Japan, surface water is regarded as a public resource and the user is subject to various regulations, including permission requirements. However, groundwater is associated with land ownership and is not regulated by any nationwide permission system [21].

These loose regulations governing groundwater use have resulted in several problems. A typical example is the excessive exploitation of groundwater by the industrial sector during the era of economic development (1950s to 1970s) that followed the Second World War. The countermeasures against this problem comprise two main classifications, legal regulation by the Industrial Water Law of 1956 [22] and the establishment of the CCGP.

The Industrial Water Law of 1956 is a national-level law, under which the Japanese government first established "designated areas", and anyone wishing to build a new well with a cross-section area exceeded 6 cm$^2$ was required to obtain permission from the government of the prefecture in which the well was to be located. Area designation required several conditions. First, social problems attributed to groundwater pumping (e.g., land subsidence) had to already have occurred. Secondly, there had

to be a strong need to conserve groundwater for industrial uses. Third, industrial waterworks had to already have been constructed or be due to begin construction within a year. Areas that typically satisfied these conditions were large cities, such as Tokyo and Osaka. However, as the first condition detailed above suggests, this law was not preventive [23].

Japan's Ministry of International Trade and Industry (MITI) was in charge of industrial water supply when the Industrial Water Law was enacted, and took several preventive measures, foreseeing that the development of groundwater for industrial use would spread to local areas. The MITI selected locations where demand for groundwater for industrial uses was very large and helped local governments promote reasonable use of local groundwater by conducting scientific research on geological conditions and groundwater flow. The first study was conducted in 1965 and 121 locations have hitherto been surveyed. Among them were some places where local governments and groundwater users had established organizations (i.e., CCGPs) under the instruction of the MITI, to promote groundwater protection [24].

While CCGPs initially acted individually, seven established an association in November 1969, for the primary purpose of sharing information. The CCGPs of other areas joined later, and it became a nationwide association in June 1976 [24]. Table 1 shows the 25 CCGPs that are members of the national association as of July 2017. The number of members fluctuates as members join or exit the association [25]. The years in parentheses indicate the year of each CCGP's establishment. Among the current 25 CCGP members, the majority were launched in the 1960s and 1970s. As mentioned above, the Industrial Water Law was enacted in 1956. This implies that the law was first implemented where excessive groundwater pumping was an ongoing reality and the CCGPs were later established in other areas.

**Table 1.** Councils for Coordinated Groundwater Pumping as of July 2017.

| No. | Location of the Councils | No. | Location of the Councils |
|---|---|---|---|
| 1 | Kesen-numa area, Miyagi Prefecture (1983) | 14 | Gakunan area, Shizuoka Prefecture (1967) |
| 2 | Higashi-ne area, Yamagata Prefecture (1978) | 15 | Seishin area, Shizuoka Prefecture (1976) |
| 3 | Tendou area, Yamagata Prefecture (1978) | 16 | Oigawa area, Shizuoka Prefecture (1969) |
| 4 | Yamagata area, Yamagata Prefecture (1976) | 17 | Chu-en area, Shizuoka Prefecture (1972) |
| 5 | Yonezawa area, Yamagata Prefecture (1976) | 18 | Sei-en area, Shizuoka Prefecture (1971) |
| 6 | Jouetsu area, Niigata Prefecture (1963) | 19 | West Hamana lake area, Shizuoka Prefecture (1979) |
| 7 | Kurobegawa area, Toyama Prefecture (1991) | 20 | Mukou City, Kyoto Prefecture (1992) |
| 8 | Uozu and Namerikawa areas, Toyama Prefecture (1989) | 21 | Nagaoka-kyo area, Kyoto Prefecture (1982) |
| 9 | Toyama area, Toyama Prefecture (1975) | 22 | Touban area, Hyogo Prefecture (1968) |
| 10 | Shougawa and Oyabe area, Toyama Prefecture (1987) | 23 | Lower Yoshino River area, Tokushima Prefecture (1969) |
| 11 | Gifu area, Gifu Prefecture (1975) | 24 | Central Kagawa area, Kagawa Prefecture (1981) |
| 12 | Seinou area, Gifu Prefecture (1974) | 25 | Saijo City, Ehime Prefecture (1973) |
| 13 | Kisegawa area, Shizuoka Prefecture (1974) | | |

### 3.2. Spatial Distribution of CCGP

Figure 1 illustrates the distribution of designated areas under the Industrial Water Law and the locations of CCGPs. Prefectures that include designated areas in their jurisdictions are meshed in the figure. There were 17 designated areas in 10 prefectures as of 2016 [26]. The locations of CCGPs are numbered (1–25) to correspond with the numbers in Table 1. Figure 1 shows that the CCGP locations do

not overlap across the 10 prefectures mentioned above. (CCGP No. 22 is located in Hyogo Prefecture meshed in Figure 1. While Hyogo's CCGP is located in Takasago City, the prefecture's designated areas are the cities Amagasaki, Nishinomiya, and Itami, and thus there is no overlap.)

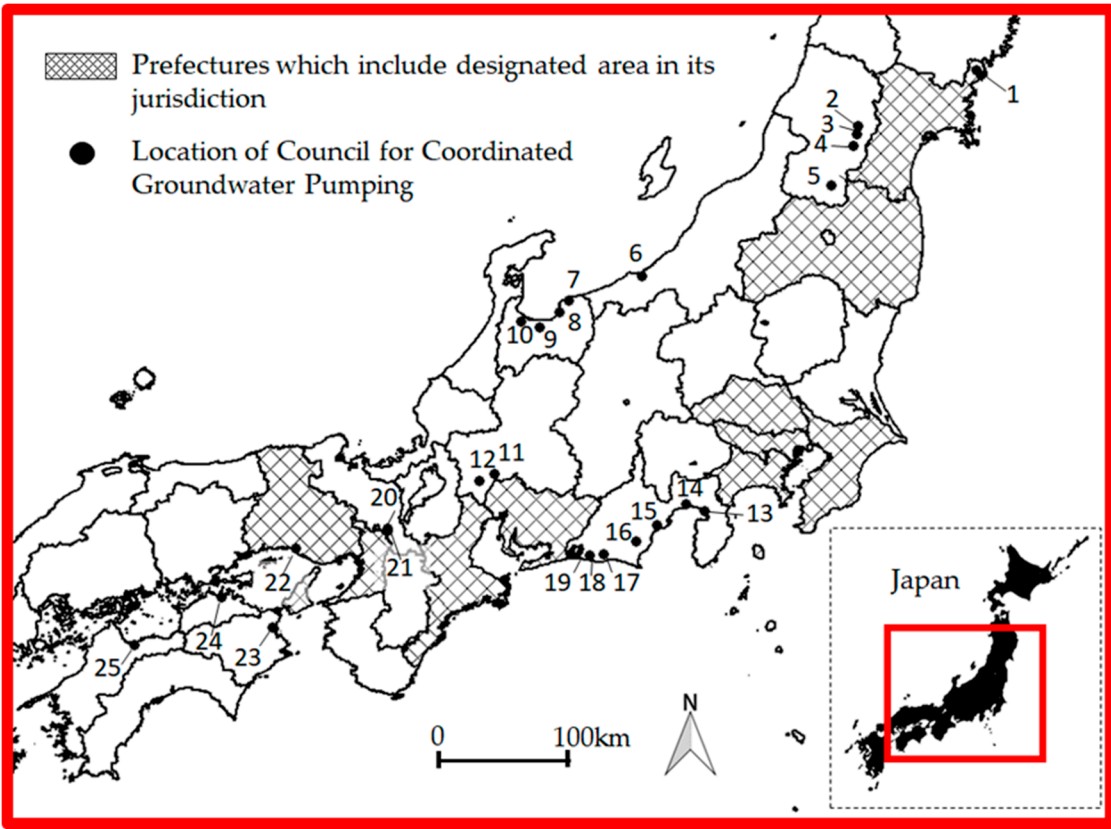

**Figure 1.** Location of Councils for Coordinated Groundwater Pumping and prefectures that have designated areas based on the Industrial Water Law of 1956.

As mentioned above, studies of the groundwater utilization conditions were conducted by MITI in 121 locations between 1965 and 2007. The first research was carried out in Fuji City, Shizuoka Prefecture, and Gakunan CCGP was subsequently established in 1967. This is a pioneering case that served as a model for other CCGPs established later [27]. Generally speaking, a common problem faced by CCGPs across Japan was excessive groundwater pumping, although some CCGPs, including Fuji City, also suffered secondary problems such as seawater intrusion. The next chapter offers a brief history of Gakunan CCGP and an exposition of its institutional structure, by focusing on the following four key elements of groundwater governance identified by Varady et al. (2016) [12]: Availability of and access to information and science, institutional setting, robustness of civil society, and economic and regulatory framework.

## 4. Gakunan Council for Coordinated Groundwater Pumping

### 4.1. Groundwater Issues in the Gakunan Area

The Gakunan area is located in southern Shizuoka Prefecture. "Gaku" is generally best interpreted as another expression for a large mountain in Japanese. Here, it denotes Mt. Fuji, Japan's highest mountain. The "nan" constituent of the name denotes the south. Therefore, Gakunan can be translated as the area to the south of Mt. Fuji [28]. Gakunan's main urban area is Fuji City, which is located in the furthest downstream region of the Fuji, Urui, Taki, Akabuchi, and Numa rivers. The city is also rich in groundwater, which is recharged in the southern slopes of Mt. Fuji (Figure 2).

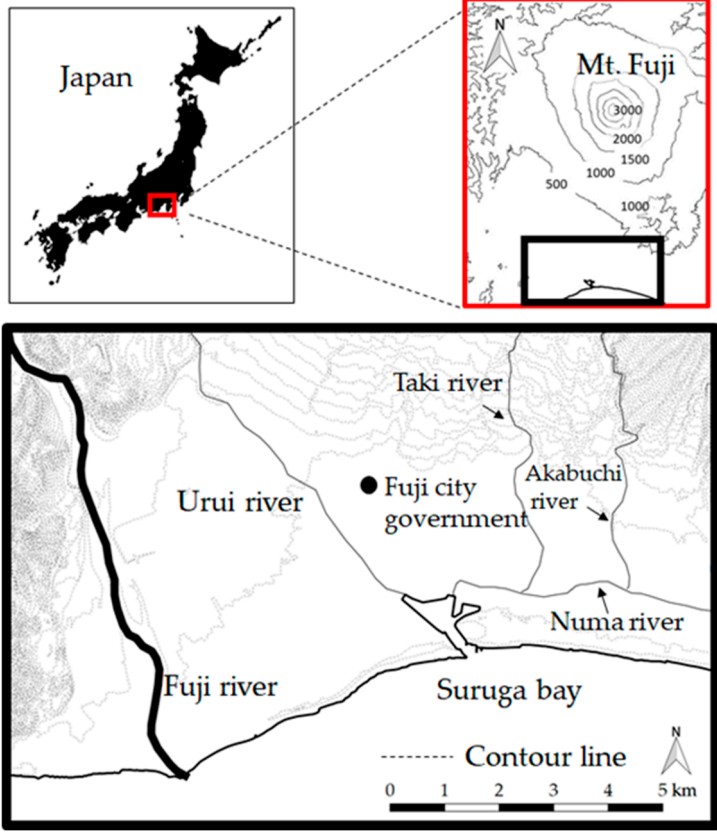

**Figure 2.** Map of Gakunan area.

Fuji City is one of the major centers of Japan's paper industry. Whether paper is made from pulp or used paper, large amounts of water are necessary in either case, to facilitate the fiber disintegration and papermaking processes. Rich groundwater was a primary reason for the establishment of several paper companies in Fuji City.

During the mid-1950s, a survey on water utilization in Fuji City was conducted by the Geological Survey of Japan, which is a national research institute operating under MITI. Groundwater was the main water source for industrial purposes, particularly the paper industry; it was also used for domestic use, but in a far smaller volume [29]. However, massive groundwater pumping precipitated seawater intrusion in the city's coastal area. Seawater intrusion was first observed in 1960; it imposed negative effects on paper production machinery and compromised the quality of finished products. Several companies sought to overcome this problem by digging deeper wells in areas further inland, but the seawater soon began to encroach on these wells. Groundwater users, thus, began to realize the necessity of countermeasures [30,31].

### 4.2. Availability and Access to Information and Science

In 1965, MITI and the Geological Survey of Japan initiated a survey that had two objectives, to assess the status of industrial water usage and to formulate a plan that would facilitate both the use and protection of groundwater. Yamazaki and Murashita (1966) [28] detailed the research method and the results as follows:

A questionnaire on industrial water use was distributed to local factories with over 30 employees. Overall, 224 factories satisfied the criteria for inclusion. These comprised not only paper companies but also food-processing and chemical companies. On the basis of the survey's findings, the volumes of industrial use per day and per factory were calculated by dividing the volume of groundwater use between January 1 to December 31 of 1964 by the number of operational days. Where the volume

of groundwater pumped was uncertain, it was estimated based on the pump power and the length of operation.

Of the 224 factories, 172 responded to the questionnaire. The results were that industrial water use per day was around 1.7 million m$^3$, and 79.4% of the total use (1.35 million m$^3$ per day) was supplied by groundwater, while the rest comprised surface water, recycled water, and industrial waterworks. This questionnaire verified that industries in Fuji City were highly dependent on local groundwater (Figure 3).

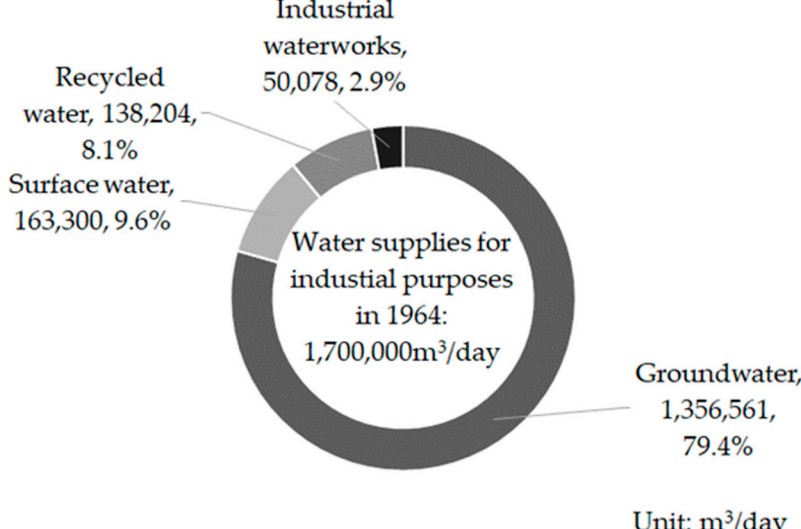

**Figure 3.** Water supplies for industrial purposes in 1964.

Research was also carried out in relation to geological conditions, groundwater levels, and hydraulic conductivity, and the upper limit of groundwater pumping was estimated to be 0.89 million m$^3$ per day. Here, the upper limit denoted the maximum amount of groundwater pumping that would not cause seawater intrusion and land subsidence. (Later, following the subtraction of 0.09 million m$^3$ per day for future drinking use, 0.8 million m$^3$ per day was established as the revised upper limit.) As mentioned above, the survey of industrial water use revealed that the total volume of groundwater pumped amounted to 1.35 million m$^3$ per day. It was, thereby, confirmed that groundwater use exceeded the upper limit [28,32].

*4.3. Robustness of Civil Society*

MITI and the prefectural government commenced discussions regarding how groundwater pumping might be reduced. The following three countermeasures were considered: voluntary reduction on the part of factories; reduction with the cooperation of factories and the city, prefectural, and national governments; and implementation of the Industrial Water Law. The proposal of voluntary reduction was rejected on the grounds that this would be difficult to coordinate, considering that there were numerous factories of various sizes. The implementation of the Industrial Water Law enabled prefectural government to regulate new wells and abolish existing ones, however, there was some anxiety that such strong regulation might impose negative effects on the local economy, which depended considerably on access to groundwater. Consequently, the cooperative groundwater management strategy was selected, due to its predicted efficacy and the possibility of its swift implementation [30,33].

Gakunan CCGP was established in 1967, with the objective of putting an end to excessive pumping and protecting local groundwater resources through cooperation between the private and public sectors. The organization comprises 32 representatives from various sectors, i.e., the national government (two), Shizuoka prefectural government (one), Fuji City government (two), commerce association of Fuji City

(four), paper companies (17), and other private companies (six). Furthermore, private companies that use more than 500 m$^3$ groundwater per day (120 companies) or that intended to begin operating in Fuji City in the near future were also registered as members of Gakunan CCGP. [30].

In time, Gakunan CCGP gradually became a transboundary organization, inviting groundwater users from adjacent cities, i.e., 129 companies in Fuji, 57 in Fujinomiya, and 14 in Shizuoka were registered members of Gakunan CCGP as of 2015 [34].

*4.4. Institutional Settings*

Gakunan CCGP determined the organization's constitutional rules and the basic policy for groundwater control. The former delineated the organization's purpose, functions, structure, decision-making guidelines, and financial sources, while the latter set the following subset targets: to end the expansion of groundwater pumping to avoid further seawater intrusion, to promote water recycling to cope with increasing demand, to construct the East Suruga Bay industrial waterworks, and to formulate a plan for the exploitation of deeper groundwater resources [30].

Gakunan CCGP contacted not only existing members but also newcomers. The organization required its existing members to submit information with regard to the location, depth, and pumping volume per day of each well in existence. The CCGP also established new technical standards and requested that newcomers comply with them. The standards governed the depths of strainers, the distance between wells, and the maximum amount of groundwater pumping permitted per day. Those planning to use groundwater were required to register in advance, submitting information regarding their purposes in using groundwater, the locations and depths of the wells, and the pumping volume per day. While responding to this request was not mandatory, it was nonetheless an important measure in the restriction of new wells [27,35].

Securing financial resources is a major challenge in the promotion of groundwater governance [4,10]. In the case of Gakunan CCGP, the budget was initially financed in its entirety by Shizuoka Prefecture and Fuji City governments, and the member companies were later made liable for fees. The fee is collected once a year and comprises two parts. First, each member is required to pay JPY 1000 (approximately 9 USD) as a basic fee, which is fixed, irrespective of the volume of groundwater pumped. Second, an additional fee is owed according to the pumped volume. The prices are categorized into 10 block levels and the payment increases proportionally to the volume of groundwater pumped. Table 2 presents the block fee structure as of 2015 [34].

**Table 2.** Structure of block fee.

| Levels | Volume of Groundwater Pumping (m$^3$/day) | Fee (JPY) | Fee (USD) |
|--------|------------------------------------------|-----------|-----------|
| 1 | 200,000– | 40,000 | 364 |
| 2 | 100,000–199,999 | 30,000 | 273 |
| 3 | 50,000–99,000 | 20,000 | 182 |
| 4 | 30,000–49,999 | 15,000 | 136 |
| 5 | 10,000–29,999 | 10,000 | 91 |
| 6 | 5000–9999 | 5000 | 45 |
| 7 | 3000–4999 | 3000 | 27 |
| 8 | 1000–2999 | 2000 | 18 |
| 9 | 300–999 | 1000 | 9 |
| 10 | –300 | 0 | 0 |

(1USD = 110JPY).

According to a member company of Gakunan CCGP, the additional fee is determined by the expected total amount of pumped volume (m$^3$/day), (which is registered to Fuji city government, a secretariat of Gakunan CCGP) in advance. Active wells and dormant wells are taken into consideration. For example, if a company has active wells for which the expected pumped volume is 5000 m$^3$/day and dormant wells for which the registered pumped volume is 6000 m$^3$/day, respectively (11,000 m$^3$/day in total), the company is required to pay an additional fee that corresponds to the volume of level 5 in Table 2, that is, JPY 10,000 (approximately 91 USD). It is difficult for each company to pump more groundwater than the registered volume because pumps have been equipped with control devices so that pumping does not exceed the registered volume. These measures prevent dishonesty [36].

Thus, Gakunan CCGP is now financed by member fees in addition to the government subsidy. These funds are primarily directed toward monitoring groundwater and seawater intrusion levels. As of 2016, there were 19 wells for monitoring groundwater levels in Fuji; of these, 11 are managed by Gakunan CCGP while the rest are managed by the prefecture and city governments. Gakunan CCGP has 46 wells for monitoring seawater intrusion and has sustained its observational activities for 50 years. [34].

### 4.5. Economic and Regulatory Framework

Industrial waterworks supported by surface water played a major role in reducing groundwater pumping. Figure 4 illustrates the locations of the main paper factories in 1955 [29] and the pipelines of contemporaneous industrial waterworks [37]. As can be seen from the illustration, groundwater pumping was particularly active in the Tadehara, Denpoh, and Imaizumi areas of Fuji City. The Fuji River Industrial Waterworks were constructed in 1964 and the East Suruga Bay Industrial Waterworks were constructed in 1971 by the prefectural government, promoting reassignment of the water supply from groundwater to surface water. The arrows in Figure 4 indicate the flow directions of industrial waterworks.

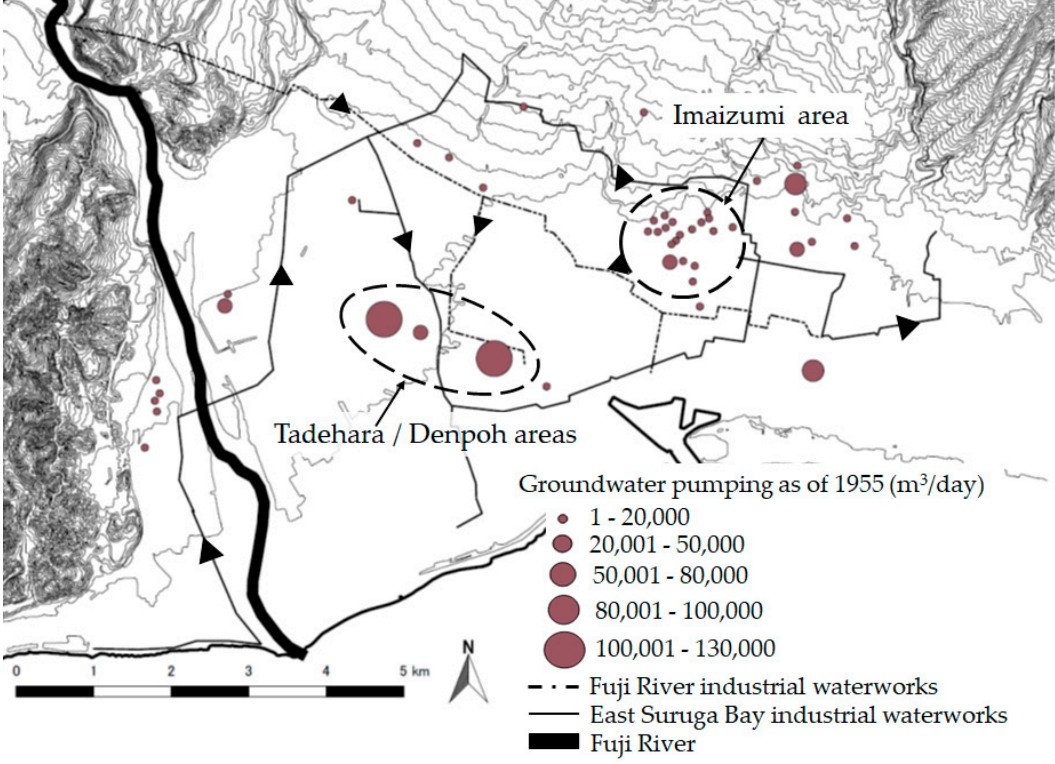

**Figure 4.** Groundwater pumping by paper companies as of 1955 and current industrial waterworks, in the Gakunan area.

Although the reassignment of the water supply to surface water (industrial waterworks) was considered a promising solution, it was difficult to ensure the voluntary cooperation of factories. There were several reasons for this lack of voluntary cooperation. The price of groundwater (2 JPY per m$^3$) was much cheaper than that of surface water (7 JPY per m$^3$) at the time. Moreover, the quality of the surface water was inferior to that of the groundwater. Finally, the pipes were not large, and thus it was difficult for all the groundwater-based factories to reassign their water supplies simultaneously. The earlier a factory changed its water supply to surface water, the longer it was obliged to pay a higher price for lower-quality water. As such, there was little incentive for factories to reassign their water supplies.

This problem was addressed by what is called a "joint-payment system." Under this system, factories were classified into two groups; the first comprised factories in a coastal area that faced severe seawater intrusion and the second comprised factories that had experienced less damage. The first group began to reassign its water supply and the second paid a subsidy. [32,38].

Nevertheless, these countermeasures implemented by Gakunan CCGP were insufficient. It is true that the registration system and the introduction of technical standards were instrumental in ending unrestricted groundwater pumping to a certain extent, but these were based on the voluntary cooperation of factories. Gakunan CCGP did not have the authority to impose their requests on their members, let alone factories who were not members of the CCGP. There were, indeed, some factories that refused to reassign their water supplies or continued to use groundwater even after the industrial waterworks had become operational [32].

The intervention of the prefectural government solved this problem. Shizuoka's prefectural assembly enacted the Prefectural Ordinance on Groundwater Pumping in Shizuoka in 1971. This ordinance was amended in its entirety by the Prefectural Ordinance on Groundwater Pumping in Shizuoka, in 1977 [39,40]. The ordinance of 1971 legalized the Gakunan CCGP, although it did not provide for punishment for the violation of the technical standards of groundwater pumping. Taking these shortcomings into consideration, the ordinance of 1978 gave the prefectural government the authority to order the temporary stoppage of groundwater pumping in the case of users who failed to comply with the standards. This reinforced punishment system supported the reduction of groundwater pumping [32,41].

*4.6. Effects of Groundwater Governance*

Figure 5 illustrates how governance worked to solve the problem of excessive groundwater pumping [34,42,43]. The upper part of the illustration describes the relationship between the upper limit of groundwater pumping, actual groundwater pumping, and groundwater levels. As mentioned above, groundwater had been extracted above the upper limit prior to the early 1970s. However, water supply reassignment by the industrial waterworks reduced the volume of groundwater pumped to below the limit, resulting in the restoration of groundwater levels.

The lower part of the illustration shows how the water supply changed over time. The substitutes for groundwater were industrial waterworks (surface water) and recycled water. The volumes of both water supplies were almost equivalent initially, but the share of recycled water gradually increased.

The expansion of water recycling can be partly attributed to the construction of a sewage system for the industrial water. Paper companies tended to discard the remnants of paper production into adjacent rivers, creating problems for the agricultural and fishery sectors. Consequently, the construction of sewage plants specific to the industrial sector commenced in 1951, and the recycling area covered gradually expanded. The Gakunan Association for Management of Drainage Water (GAMDW) has been in charge of these facilities. Each factory is required to install a meter authorized by GAMDW. The volume of drainage water is gauged once a month by GAMDW staff to prevent dishonesty. A fee is collected every month. The total fee comprises a basic fee, which is fixed irrespective of the volume of drainage water released, and an additional fee, which is in proportion to the volume. This has allowed

factories to use water repeatedly during production to reduce drainage water, which has resulted in decreased demand for groundwater [44,45].

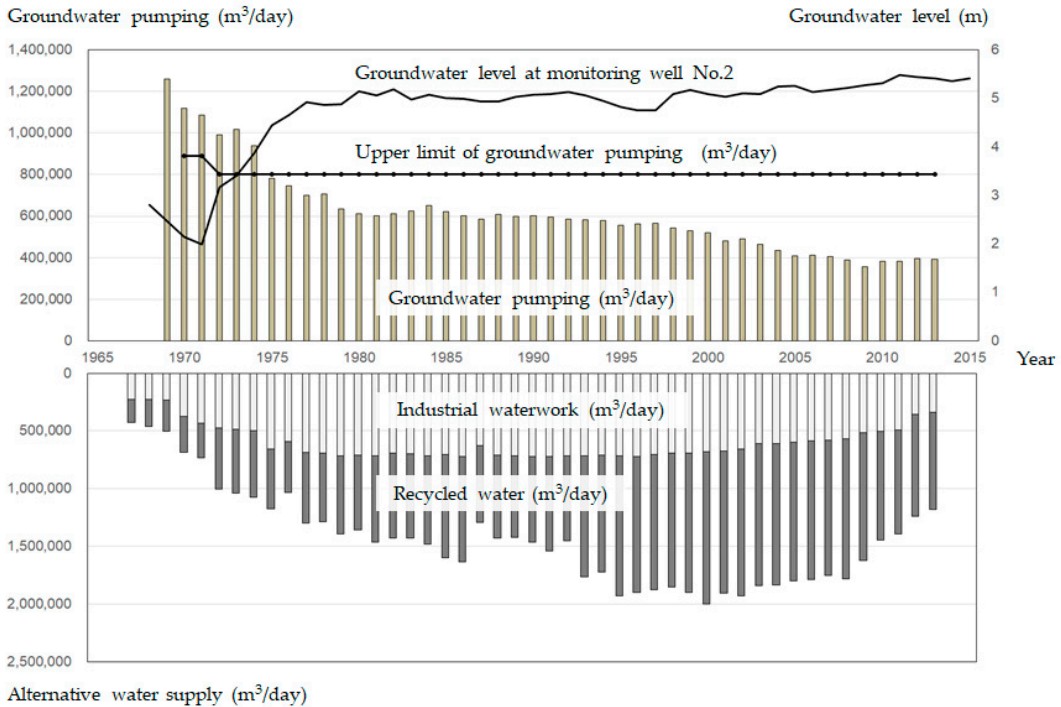

**Figure 5.** Effects of groundwater governance in the Gakunan area.

Attention has increasingly focused on the adoption of an indirect approach to the control of uncoordinated groundwater pumping. Shah et al. (2012) [46] reported a case in India where agricultural groundwater pumping was reduced by reducing the subsidy for the electricity used to pump the groundwater rather than by direct regulation. The Gakunan CCGP case can be considered a similar example because the charge for the volume of water disposed to a sewage plant was instrumental in restricting groundwater use. The effects of such a charge on groundwater pumping have also been described by Kataoka [47].

## 5. Analysis by Groundwater Governance Matrix

Securing the participation of various stakeholders and establishing the appropriate division of labor are key elements in groundwater governance [8,10,11]. In this section, some observations arising from these issues are considered.

### 5.1. Groundwater Governance Matrix

Studies of groundwater governance have focused on the process from the initial recognition of the problem to its eventual solution. How various stakeholders use diverse tools to address the problem and to what extent these tools are appropriate are also issues of significant concern. Varady et al. (2013) and De Chaisemartin et al. (2017) [3,9] have already proposed an analytical framework for this process, naming it the "development stages of groundwater management". They classified the stages into the following three stages: the pre-management stage, the initial (crisis) stage, and the advanced management stage. During the pre-management stage, there are no institutions aimed at coordinating groundwater pumping and free competition for groundwater is possible. The initial (crisis) management stage begins when a groundwater issue and the necessity to find a solution are recognized; countermeasures taken at this stage are ex post facto and emergent ones. The advanced

management stage begins when groundwater pumping is controlled along with other related sectors, such as land use. At this stage, preventive measures become the prevalent issue.

Although this stage classification is reasonable, there has been little exploration as to how it can be applied to the study of groundwater governance. To link the stage classifications to groundwater governance, the key actors and their policy tools at each stage should be considered. For the purpose of this paper, we term the correspondence between the actors and each stage the "groundwater governance matrix." It is worth delineating the structure of groundwater governance under consideration. Taking Gakunan CCGP as an example, the groundwater governance matrix is detailed in Table 3.

**Table 3.** Governance matrix in the Gakunan area.

| Year | 1960 | 1967 | 1970 |
|---|---|---|---|
| Stage | Pre-management | Initial (crisis) management | Advanced management |
| Phenomena | | Sea-water intrusion | Groundwater level recovery / Decrease in seawater intrusion area |
| Users | Free groundwater pumping | | Self-regulation/conversion of water supply / water recycling/monitoring → |
| City | | | Formation of Gakunan CCGU / Secretariat function → |
| Prefecture | | | Industrial waterworks / Prefecture ordinance → |
| National government | Analysis on groundwater uses / Monitoring → | | |

Among the major challenges to efficient groundwater governance is enticing the various stakeholders to participate and appropriately allocating functions to them [8,10,11]. As De Chaisemartin et al. (2017) pointed out, stakeholder participation remains poorly developed in several cases [9]. This prompted us to ask what made active participation successful in the case of Gakunan CCGP.

Shared policy goals centered on monitoring are considered to have been one reason for the organization's success. As mentioned above, the upper limit of groundwater pumping (0.89 million $m^3$ per day) was detailed by the Study on Reasonable Groundwater Use carried out by MITI. Following the subtraction of 0.09 million $m^3$ per day for future drinking use, 0.8 million $m^3$ per day was established as the revised upper limit. It was assumed the groundwater pumping had exceeded the limit since 1960s (see Figure 5), and thus the industrial sector was required to reassign the difference to other sources. This information was shared with the stakeholders and countermeasures were taken from the ground up.

Ross and Martinez-Santos (2010) reported that the establishment of a collaborative framework among the various stakeholders remained a major challenge [48]. The optimal approach to shaping a productive relationship between private groundwater users and the government has not yet been identified and requires further investigation. The case of Gakunan CCGP offers a potential solution to this issue, i.e., cooperation between the public and private sectors will be fostered when the government adopts a more neutral approach to its intervention, such as the sharing of information rather than coercive measures, such as punishment for excessive groundwater use. Moreover, the case

of Gakunan CCGP also suggests that this neutral approach should be implemented from the earliest stages of intervention.

Needless to say, shared goals will not always engender cooperation among stakeholders. As Olson (1965) demonstrated in his seminal work, shared goals can cause a free-rider problem, whereby stakeholders expect contributions from others without investing any effort of their own [49].

Gakunan CCGP managed to avoid a free-rider scenario because of three key factors, which could be specific to local conditions. The first factor was the region's geographical setting. The plain area in which the factories were concentrated was not large and was surrounded by mountains and sea. This area limitation restricted factory numbers. The smaller the group size, the easier it is to elicit voluntary cooperation [49,50]. Second, the "joint-payment system" mentioned above played a role. The free-rider scenario is more likely to occur when the difference between an individual party's benefits and costs from a shared goal become large [18]. The payment system averaged out costs to reassign its water supply, thereby contributing to a solution to the free-rider problem. Third, representative democracy governed Gakunan CCGP's decision-making culture. This allowed a limited number of representatives to enter into discussion with one another, thereby further reducing the transaction cost.

The second factor involved the users' characteristics. The primary groundwater users were paper companies who were highly dependent on groundwater for production. Both the quality and the quantity of groundwater was of major concern to them because seawater and surface water which have been contaminated by domestic wastewater cause irregular coloration of the finished products. As such, groundwater was indispensable to the production process, an integral ingredient for which substitutes were not easily found, and therefore factories had a strong incentive for its protection.

### 5.2. Importance of Metagovernor

Encouraging the participation of various stakeholders poses a challenge, and coordination of their activities also presents several problems. As detailed below, the governance matrix indicates that a bottomex-up approach is the optimal strategy for effective coordination.

In hindsight, in implementing countermeasures aimed at tackling seawater intrusion, the central government (MITI) first raised public awareness regarding this issue by sharing monitoring information. This prompted the establishment of Gakunan CCGP, which introduced a set of technical standards for groundwater pumping. While this was instrumental in ending uncoordinated groundwater production to some extent, it was insufficient. As additional countermeasures, the members of Gakunan CCGP voluntarily promoted the reassignment of the water supply from groundwater to the industrial waterworks. The joint-payment system was introduced to address any contention regarding who had to reassign their water supply first. However, what Gakunan CCGP actually promoted was voluntary reduction of groundwater pumping and the organization was not granted the authority to punish those failing to abide by the technical standards, nor did it have the authority to coerce nonmembers to join. Ultimately, to overcome this problem, the prefectural government enacted a new local ordinance that bestowed a defined legal status on Gakunan CCGP and introduced a penalty clause to deal with offenders. The prefectural government simultaneously expanded the industrial waterworks to assist the water-supply reassignment process. This sequence of events attests that the various policies were shaped by a bottom-up approach.

The role of the Fuji City government is particularly noteworthy in this case. The city government functioned as the secretariat of Gakunan CCGP and formed a hub connecting the various stakeholders with one another. Therefore, contradictory actions on the part of individual stakeholders could be avoided due to the centralization provided by this hub. An actor whose role is to coordinate the various other stakeholders is termed a "metagovernor" [51]. Fuji City fulfilled this very role in this example, resolving the problem of coordinating the activities of the various stakeholders, which is a significant issue in governance studies in general.

Although studies of groundwater governance emphasize the importance of stakeholder participation, in reality, it is difficult to make contact with stakeholders and persuade them to

join a collaborative endeavor aimed at the protection of groundwater within a short period. To address this problem, an incremental approach was proposed. Collaboration between stakeholders is gradually realized through a trial-and-error process of incrementalism. It is asserted that open-access information and environmental assessment are key factors for the promotion of incrementalism [52]. While this is a valid argument, the case of Gakunan CCGP indicates that the presence of a metagovernor as a centralizing force is an important additional factor.

### 5.3. Significance of Groundwater Governance

As research on groundwater governance has increased, several definitions of the concept have been proposed. Despite the variety of definitions, Mukherji and Shah (2005) demonstrated that they share some common denominators, i.e., groundwater governance is a means of conserving groundwater that is implemented by multiple stakeholders from multiple levels with multiple instruments [2].

Although numerous studies have sought to define the concept, little attention has been paid to the necessity of groundwater governance. While some studies have offered justification for the concept, they have tended to imply a tacit assumption that groundwater protection through stakeholders' collaboration is the most desirable approach. It is true that stakeholder participation may contribute to the creation of localized, tailor-made solutions, but it may also entail time-consuming negotiation processes. This prompts us to ask, what makes groundwater governance so important in the face of challenges such as these.

The case of Gakunan CCGP indicates that its significance lies in the fact that groundwater governance pools various tools and instruments for groundwater protection and enables stakeholders to use them appropriately in accordance with the circumstances. As Theefeld (2009) summarized, instruments for groundwater protection are classified into three groups, regulatory policy instruments, economic policy instruments, and voluntary policy instruments. Regulatory policy instruments are based on government coercion and include protection of groundwater rights and pumping restrictions. Economic policy instruments make use of market principles and pump taxes and subsidies for water-saving measures are typical examples. Voluntary policy instruments motivate voluntary action without financial incentives [20].

On the basis of these classifications, the countermeasures against excessive groundwater pumping mentioned above may be explained as follows: The registration of wells by Gakunan CCGP corresponds with the voluntary policy instruments at user-level. Water-supply reassignment was promoted by the joint-payment system, which introduced internal subsidies among the members. This is considered an economic policy instrument. However, as soon as it emerged that voluntary and economic policy instruments were insufficient, the prefectural government intervened to enact a local ordinance aimed at further reduction of groundwater pumping. This corresponds with the regulatory policy instruments wielded by higher-level authorities.

As this example demonstrates, various policy instruments were implemented by various actors at different levels as the process continued. Lopez-Gunn (2009) observed that groundwater governance is desirable, in that it broadens the range of policy options [11]. However, its real significance does not lie in the expansion of policy instruments, but rather in the interconnection of these instruments toward a locally shared goal.

## 6. Conclusions

As stated, this paper has two objectives. First, to clarify the history and functions of Gakunan CCGP with reference to governance studies and, second, to extract general lessons for future groundwater governance from the case study.

Gakunan CCGP is an organization whose purpose is to prevent excessive groundwater pumping through the cooperation of both the public and private sectors. Its function is to connect multiple actors from multiple levels, including groundwater users and the city, prefectural and national governments,

and to use multiple instruments to this end appropriately, including voluntary regulation, internal subsidies between groundwater users, and legal regulations from the upper levels of the government.

Several lessons were extracted from the Gakunan CCGP case study using a governance matrix, which illustrates the correspondence between the developmental stages of groundwater management and the primary actors at each stage. First, the participation of the stakeholders was fostered by shared goals associated with the monitoring conducted by the central government. This implies that the earlier neutral interventions, such as the communication of information, are implemented, the better the outcome will be. Second, this paper highlights the significance of a metagovernor that fulfils a hub-type function among the stakeholders. More important than encouraging the participation of various stakeholders is the meticulous coordination of their activities. Finally, this study considered the aspects that make groundwater governance a necessity. This paper proposes that it creates a pool of various policy instruments that enable stakeholders not only to expand the range of policy options but also to implement them appropriately with due consideration of the circumstances.

Two key topics remain to be scrutinized. In the case of Gakunan CCGP, the primary user of groundwater is the industrial sector. Hitherto, most studies of groundwater governance have focused on groundwater production in the agricultural sector; comparative studies of groundwater governance as it relates to both the agricultural and industrial sectors are recommended. The numbers and area distribution of stakeholders and the year-round pumping patterns vary across the two sectors and may influence the structure that groundwater governance adopts. Second, the environment should be incorporated as a stakeholder in future research. Governance studies emphasize the importance of participation from stakeholders of various backgrounds, tacitly limiting the definition of stakeholder to people. We need only look to wetlands to understand that groundwater is also a crucial constituent of the ecosystem. Future studies should investigate how demand from the environmental sector might be reflected in groundwater governance. Groundwater governance that also prompts the establishment of ecosystem services is recommended as an important avenue for future exploration.

**Funding:** This research was supported by the research project "Fusion of global water resources assessment and social institution studies" (Director Shinjiro Kanae, Professor, Tokyo Institute of Technology), under the Japan Society for the Promotion of Science KAKENHI (Grant-in-Aid for Scientific Research (B) grant number 15H04047), "Research on the possibility of social consensus for the protection of the water environment" donated by the Osaka branch of Yachiyo Engineering CO., LTD, and a donation by the Institute for Water Science, Suntory Global Innovation Center Limited.

**Acknowledgments:** We acknowledge helpful comments from Shinichi Yatsuki, Tomoyo Chiba, and anonymous reviewers. Special thanks are due to the Department of Industrial Policy, Fuji City government and Gakunan area branch, the east office, the public enterprise agency, Shizuoka Prefecture for providing the data, and to S. Takamori, (Yachiyo Engineering CO., LTD), N. Yamada, T. Yamada (a staff of a member company of Gakunan CCGP) and M. Kubota (Shizuoka Prefectural Technological Association on Paper and Pulp) for research assistance. The views presented here are those of the author and should in no way be attributed to any organization or individual mentioned above. Finally, we thank the reviewers for their useful comments. Responsibility for the text rests entirely with the author. We thank Textchek (http://www.textcheck.com) for editing a draft of this manuscript.

**Conflicts of Interest:** The authors declare no conflict of interest.

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
