# Peer review of "A Fifty-Year Experience of Groundwater Governance: The Case Study of Gakunan Council for Coordinated Groundwater Pumping, Fuji City, Shizuoka Prefecture, Japan"

_water, doi:10.3390/w11122479_

Round 1

Reviewer 1 Report

I have a few recommendations for the authors to improve the ability of readers to access and interact with the paper once it is published.

Put a bit more information in the abstract and introduction on what the paper does and what it’s key findings are.

In particular, the problem: “massive groundwater pumping precipitated seawater intrusion in the city’s coastal area” due to low cost of extraction and incomplete groundwater ownership (common-pool problem)

The solution: Legal authority to regulate wells from the industrial water law coupled with the adoption of a cooperative groundwater management strategy to reduce pumping and coordinate surface water financing and delivery and the outcome: increased elevation of groundwater tables and the keys to success: the legal authority to regulate groundwater; investment in alternative water supplies; the ability to charge for water disposal; small group size;

“cooperation between the public and private sectors…sharing of information rather than coercive measures.”

Are not clearly articulated in the abstract or introduction.

Add in the conclusion more on the limitations of the work and how they agree or diff. For instance, you suggest that the lack of prescriptive regulation was an important aspect. Yet the lack of prescriptive regulation

Author Response

Dear reviewer,

Thank you very much for your comment. Following your comment, I modified abstract. Please check attached file.

Reviewer 2 Report

This manuscript is a resubmission of the manuscript with id water-574199. The author has addressed all the comments of the previous review.

Author Response

Dear reviewers,

Thank you very much. Your comments was really helpful.

Sincerely yours,

T.Endo

This manuscript is a resubmission of an earlier submission. The following is a list of the peer review reports and author responses from that submission.

Round 1

Reviewer 1 Report

GENERAL COMMENTS

This study regards the lessons learnt after 50 years of groundwater governance coordinated by the Gakunan Council for Coordinated Groundwater Pumping. The manuscript is interesting, and well written. However, some points need further clarification and some details are missing (see specific comments below).

SPECIFIC COMMENTS

Location: "... to conserve water for groundwater protection to cope with increasing demand for industrial water."
Comment: This is not clear. The objective was to conserve water or to cope with the increasing demand?

Location: "... where large-scale groundwater development was likely to occur ..."
Comment: The meaning is not clear.

Location: " ... seven established an association in November 1969, for the primary purpose of sharing information."
Comment: This happened spontaneously? More information is required on the steps the led to the formation of this association.

Location: "... it was also used for drinking... "
Comment: Do you mean domestic use?

Location: "... used groundwater so intensively that the pumping volume per well and within a particular area of Fuji City was among the highest recorded 196 in Japan at the time [28].""
Comment: This index (pumping volume per well) does not give any information related to water management. Therefore is not useful to groundwater management and governance. This index only gives the average borehole capacity in this specific region, not the overall exploitation intensity.

Location: "... overcome this problem by digging deeper wells ..."
Comment: This does not make sense. Seawater intrusion takes place initially at greater depths, then gradually to the whole aquifer. Therefore, if water at a certain aquifer depth and location is saline, most certainly the quality will be worse at any greater depth of the same location. Unless it is a case of overlaid aquifers, an unconfined over a confined.

Location: Line 207
Comment: A paragraph cannot end with a colon.

Location: Lines 267-279
Comment: It is not made clear what is the period over which these fees apply.

Location: The prices are categorized in 10 block levels and the payment increases proportionally to the volume of groundwater pumped.
Comment: How reliable measurements of this volume are obtained?

Location: " in 1972, and found that the groundwater pumping volume in Fuji City was 1.4 million m3 per "
Comment: After 5 years of CCGP operation no reduction has been achieved? Furthermore this number is not what Fig. 5 gives for 1972.

Location: "A factory was required to pay in accordance with the volume of drainage water."
Comment: How this volume was measured. Actually, this alone would suffice to solve the problem. The challenge is to get reliable measurements of drain volume. How this was issue was resolved?

Location: "... by controlling the electricity used to pump up the groundwater."
Comment: How can be achieved a control over the various energy consumptions of a plant?

Location: "... the construction of a sewage plant was instrumental in restricting groundwater use."
Comment: This is not precise. It was the charge over the volume of disposed water that gave the incentive to recycle, not the sewage plant. The former could have been imposed without the latter. Furthermore, the plants could have kept draining to the river clandestinely. Therefore, it was the regulation and the compliance that was the key of success.

Location: Table
Comment: The row 'Phenomena' contains two terms that are not phenomena. The 'Free groundwater pumping' is a policy, therefore it should better appear as an arrow to the 'Users' row. The 'Formation of Gakuman CCGU' is a milestone, it would fit better to the row 'City'.

Location: "... smaller the group size, the easier it is to elicit voluntary cooperation"
Comment: This is a characteristic of this specific case study, not a best practice suggestion (like the representative democracy described in the next sentence).

Location: Lines 412 - 417
Comment: This factor does not explain how the "free rider" was discouraged. In other societies the majority would expect every contribution to come from the others and would evade the obligations. The success of this case must have to do with the mentality of the Japan people.

Reviewer 2 Report

Dear Author, Your topic is very timely and important. The article is very readable. You have tried to capture the steps that have been taken over the years, have been applied in the area Gakunan and have worked.

You discuss the issue of industrial customers, later extended to agriculture, and their discipline in the abstraction, consumption and recycling of groundwater. But you also point to the fact that the question is much more complicated and even depression will not prevent an undisciplined customer from doing evil.

The bottom-up management process is historically known, proven by many years and respected by all important and reasonable entrepreneurs (Tomas Bata, Emil Skoda, Salomon Reich and others only from our country). These gentlemen started out as disciples in their factories in order to understand the entire production process and thus the potential possibilities of a savings.

Undoubtedly, it is not only possible to monitor profit (individual, business), but it is necessary to take account of life and the environment as a whole. All the more in the context of today's times and changing living conditions.

Please edit the legend of Figs. 1 and 4 - hard to read,

Fig. 3 seems blurred to me.

Reviewer 3 Report

Paragraph starting on line 104 discusses difficulties of governance due to heterogeneity of parties. Might want to add the following citation:

Ayres, A.B., Edwards, E.C. and Libecap, G.D., 2018. How transaction costs obstruct collective action: The case of California's groundwater. Journal of Environmental Economics and Management, 91, pp.46-65.

Section 3: It might be useful to discuss generally the variety of groundwater problems CCGPs solve before getting into the case specifics, i.e. is the Gakunan Council reflective of the general purpose of these organizations? Or is there more variety in what problems they address.

Table 2: Are these per-unit fees or in total? Are they marginal fees or total—if someone is pumping 2000 m^3 do they pay $2000? $3000? Or some other fee?

Would like a bit more discussion on water prices—when were they implemented? Is this pricing a standard practice in CCGPs?

Would like to see a bit more discussion of the framework/matrix earlier in the paper. Especially the three groups of instruments first discussed in section 5.4 might have been introduced earlier, as they are useful for classifying types of policies throughout the paper.